# Genomic Characterization of KPC-31 and OXA-181 *Klebsiella pneumoniae* Resistant to New Generation of β-Lactam/β-Lactamase Inhibitor Combinations

**DOI:** 10.3390/antibiotics12010010

**Published:** 2022-12-21

**Authors:** Narcisa Muresu, Arcadia Del Rio, Valeria Fox, Rossana Scutari, Claudia Alteri, Bianca Maria Are, Pierpaolo Terragni, Illari Sechi, Giovanni Sotgiu, Andrea Piana

**Affiliations:** 1Department of Humanities and Social Sciences, University of Sassari, 07100 Sassari, Italy; 2Department of Biomedical Science, University of Sassari, 07100 Sassari, Italy; 3Department of Oncology and Hemato-Oncology, University of Milan, 20122 Milan, Italy; 4Multimodal Research Area, Bambino Gesù Children Hospital IRCCS, 00146 Rome, Italy; 5Hygiene Unit, Department of Medical, Surgical and Experimental Sciences, University of Sassari, 07100 Sassari, Italy; 6Department of Medicine, Surgery, and Pharmacy, University of Sassari, 07100 Sassari, Italy; 7Clinical Epidemiology and Medical Statistics Unit, Department of Medical, Surgical and Experimental Medicine, University of Sassari, 07100 Sassari, Italy

**Keywords:** carbapenem resistant Enterobacteriaceae, antimicrobial resistance, whole genome sequencing, KPC variants, porin mutations, *Klebsiella pneumoniae*

## Abstract

Background: Carbapenem resistant *Klebsiella pneumoniae* (cr-Kp) causes serious infections associated with a high mortality rate. The clinical efficacy of ceftazidime/avibactam (CZA), meropenem/vaborbactam (M/V), and imipenem/relebactam (I/R) against cr-Kp is challenged by the emergence of resistant strains, making the investigation and monitoring of the main resistance mechanisms crucial. In this study, we reported the genome characterization of a *Klebsiella pneumoniae* strain isolated from a critically ill patient and characterized by a multidrug resistant (MDR) profile, including resistance to CZA, M/V, and I/R. Methods: An antimicrobial susceptibility test (AST) was performed by an automated system and E-test and results were interpreted following the EUCAST guidelines. Genomic DNA was extracted using a genomic DNA extraction kit and it was sequenced using the Illumina Nova Seq 6000 platform. Final assembly was manually curated and carefully verified for detection of antimicrobial resistance genes, porins modifications, and virulence factors. Results: The *K. pneumoniae* isolate belonged to sequence type ST512 and harbored 23 resistance genes, conferring resistance to all antibiotic classes, including *bla*KPC-31 and *bla*OXA-181, leading to carbapenems resistance. The truncation of OmpK35 and mutation OmpK36GD were also observed. Conclusions: The genomic characterization demonstrated the high resistant profile of new cr-Kp coharboring class A and D carbapenemases. The presence of KPC-31, as well as the detection of OXA-181 and porin modifications, further limit the therapeutic options, including the novel combinations of β-lactam/β-lactamase inhibitor antibiotics in patients with severe pneumonia caused by cr-Kp.

## 1. Introduction

Antimicrobial resistance (AMR) is a major public health threat associated with increased mortality in the next decades. It has been estimated that the number of related deaths could rise to 10 million people by 2050 [1]. In 2019, drug-resistant infections caused 4.95 million deaths worldwide, surpassed only by ischemic heart disease and stroke mortality. Together, drug-resistant *Escherichia coli*, *Staphylococcus aureus, Klebsiella pneumoniae*, *Streptococcus pneumoniae*, *Acinetobacter baumannii*, and *Pseudomonas aeruginosa* were responsible for >70% of deaths [2] and, based on their role in the epidemiology of hospital-acquired infections (HAIs), they were included by the World Health Organization (WHO) in the priority list of bacteria which need urgent actions, including the development of new therapeutic options [3]. Infections caused by carbapenem-resistant *Klebsiella pneumoniae* (cr-Kp) represent one of the major concerns due to the increasing number of cases and mortality rates [2]. Since 2015 the Food and Drug Administration approved ceftazidime/avibactam (CZA), followed by meropenem/vaborbactam (M/V), and imipenem/relebactam (I/R) [4,5] for complicated intra-abdominal infections, hospital-acquired pneumonia, and urinary tract infections caused by Gram-negative bacteria producing class A, C, and D β-lactamases [6]. However, the clinical efficacy of CZA, M/V, and I/R against cr-Kp was challenged by the recent emergence of resistant strains [7]. Enzymatic modifications of antimicrobial molecules by specific variants of class A carbapenemases are associated with a significant increase in minimal inhibitory concentrations (MIC) of CZA [8,9]. Mutation within the Ω-loop of the *bla*KPC3 gene is responsible for CZA resistance in *K. pneumoniae* strains belonging to clonal complex-258 (CC258), following exposure to CZA [10]. It is essential to identify carbapenemases involved in the development of resistance, to establish the most suitable therapies. Hence, we describe the genome of a *Klebsiella pneumoniae* strain coproducing KPC-31 and OXA-181, characterized by a multidrug-resistant profile, including resistance to CZA, M/V, and I/R.

## 2. Results

A 53-year-old male was admitted to an Italian intensive care unit in 2020 for severe COVID-19-related pneumonia. No comorbidities were found and the screening for carbapenemase-producing Enterobacteriaceae was negative.

Three weeks after admission, he developed a respiratory tract infection caused by cr-Kp and died four days later after the administration of only one dose of CZA (2 g/0.5 g). 

The strain was resistant to all classes of tested antimicrobial drugs, including β-lactams, aminoglycosides, fluoroquinolones, CZA, M/V, and I/R (Table 1 and Figure 1).

Genetic analysis revealed that the strain had a genome size of 5,767,106 bp. It belonged to sequence type (ST) 512 and carried the wzi-154 capsular polysaccharide synthesis (CPS) gene, capsular locus KL107, and lipopolysaccharide O locus O2afg. Resistome analysis highlighted the presence of 23 resistance genes conferring resistance to 9 different antibiotic classes: aminoglycosides, fluoroquinolones, macrolides, chloramphenicol, rifampicin, sulfonamides, tetracyclines, trimethoprim, and carbapenems (including *bla*KPC-31 and *bla*OXA-181 genes) (Table 2 and Table 3).

The *bla*KPC31 gene was found on the Tn4401a transposon, itself carried by an IncFIB(K) plasmid. Other mobile genetic elements identified include plasmid ColKP3, carrying *bla*OXA-181 and *qnrS1* genes, and plasmid IncX3, carrying a *bla*TEM-1A gene on a Tn801 transposon.

The presence of modifications in outer membrane porin (omp) encoding genes was investigated and the truncation of OmpK35 and the mutation of OmpK36GD were observed. No genes coding for virulence factors were detected.

## 3. Discussion

Here we reported the phenotypic and genotypic characteristics of a *K. pneumoniae* strain coharboring KPC-31 and OXA-181. The expression of multiple resistance genes, together with the modification of outer membrane porins, was associated with a remarkable resistance profile to several antimicrobial drugs, including novel β-lactams/β-lactamases inhibitors CZA, M/V, and I/R. This finding is concerning, limiting therapeutic options.

The strain was found to belong to ST512, which is a variant originating from Clonal Group (CG) 258. ST512 is one of the most prevalent high-risk sequence types detected worldwide and is characterized by high transmissibility in nosocomial settings. The rapid and efficient dissemination of KPC has mostly been caused by the clonal expansion of CG 258 (e.g., ST512), as previously reported in national surveys [11,12].

The genomic characterization detected KPC-31, a KPC-3 variant characterized by the D179Y amino acid substitution. This mutation falls in the Ω-loop (amino acids 163–179) that surrounds the active site of KPC and causes a reduction of the inhibitory effect of β-lactamase inhibitors, such as avibactam [13].

Together with KPC-31, the OXA-181 carbapenemase gene was also found. After OXA-48, OXA-181 is the second most prevalent class D carbapenemase worldwide. It differs from OXA-48 because of four amino acid substitutions [14]. Further investigations on its clinical implications are required, especially for the management of infections caused by cr-Kp expressing multiple carbapenemases.

Mutations of porin-encoding genes OmpK35 and OmpK36 hinder the synthesis of functional membrane proteins and the entrance of molecules, including carbapenems, in bacterial cells [15,16]. We observed the truncation of OmpK35, resulting in a nonfunctional porin, and the mutation in OmpK36GD, which occurs near the loop 3 (L3) genetic region. The consequent effect of these alterations is the synthesis of a narrow porin which does not allow antibiotic influx [16]. Whereas OmpK36 functional alterations were described for KPC-producing *K. pneumoniae*, less is known about OXA-48-like producers, as well as their impact on antimicrobial susceptibility [17].

Although recent studies found restored carbapenem activity in CZA-resistant strains [18,19], we did not prove similar findings. Carbapenem resistance levels are affected by several factors, including the hydrolytic profile of the enzyme, gene copy number, as well as nonenzymatic mechanisms (i.e., porins modifications) [20]. We found that the cr-Kp strain showed resistance to carbapenems when tested alone or in combination with novel β-lactamase inhibitors, suggesting that the coexistence of several antimicrobial resistance genes, including both KPC-31 and OXA-181 carbapenemases, coupled with porin mutations, may be associated with an extensively-drug resistant profile.

The emergence of CZA resistance during therapy has already been described [18]. Shi et al. reported that the selective pressure of CZA favored the emergence of KPC variants with enhanced resistance [21]. Interestingly, we detected CZA resistance before antibiotic exposure. This can suggest that, even without any selective pressure, the spread of antibiotic resistance determinants may occur, often driven by mobile genetic elements.

## 4. Materials and Methods

Cr-Kp was detected in a patient admitted to the intensive care unit of an Italian university hospital with a diagnosis of COVID-19-related pneumonia. A bronchoalveolar lavage specimen was collected and used for further analysis. The automated system VITEK II [22] was used to perform strain identification and antibiotic susceptibility testing. The following antibiotics were tested: amoxicillin/clavulanate, piperacillin/tazobactam, cefepime, cefotaxime, ceftazidime, ceftolozane/tazobactam, meropenem, imipenem, amikacin, gentamicin, tobramycin, ciprofloxacin, and trimethoprim/sulfamethoxazole. Additionally, in order to perform a specific evaluation of MIC values for CZA, M/V, and I/R, Etest gradient strips (bioMérieux, Inc., Durham, NC, USA) were used. A 0.5 McFarland suspension in 0.85% sterile saline was inoculated to Mueller-Hinton agar plates with a sterile cotton swab and Etest strips were applied to plates manually. After overnight incubation at 37 °C for 16–20 h, plates were read. Results were interpreted according to the clinical breakpoints of the European Union Committee on Antimicrobial Susceptibility Testing, EUCAST v.11.0 [23]. Carbapenemases encoding genes (i.e., *bla*KPC, *bla*NDM, *bla*VIM, *bla*IMP, and *bla*OXA-48 like) were detected by real time-PCR [24].

Whole genome sequencing was performed to identify genetic determinants responsible for antimicrobial resistance and virulence. After an overnight culture of the strain, genomic DNA was extracted using a genomic DNA extraction kit (QIAamp DNA Kit, QIAGEN, Hilden, Germany) and sequenced using the Illumina Nova Seq 6000 platform (Illumina, San Diego, CA, USA). Raw reads were trimmed for adapters and filtered for quality (average quality > 20) with Fastp (v0.20.1) [25] and quality checked after trimming with FastQC (v0.11.9) [26]. A de novo genome assembly was performed using SPAdes Genome Assembler (v3.14.1) [27], using the ‘-careful’ option. The quality of the assembly was evaluated by Quast (v5.1) [28]. Annotation of the assembled contigs was performed with Prokka (v1.14.6) [29] and isolate typing was performed with the mlst tool (v2.11) [30,31]. Kaptive (v0.7.3) and Kleborate (v2.2.0) were used to predict the polysaccharide capsular (K) and lipopolysaccharide O antigen (O) loci, and antimicrobial resistance determinants and virulence factors, respectively [32,33,34]. MobileElementFinder tool (v1.0.3) [35] and PlasmidFinder (v2.0.1) [36] were used for mobile genetic elements and plasmid identification, respectively. The presence of Tn4401 transposons, their type, and flanking sequences was detected by TETyper (v1.1) [37].

## 5. Conclusions

The detection of CZA resistance in cr-Kp is a key clinical challenge: infection control measures and antimicrobial stewardship should be implemented, as well as the capacity to detect resistance mechanisms and genomic variants. Genomic analyses can help investigate the emergence of new resistance mechanisms and prevent the transmission of MDR strains.

## Figures and Tables

**Figure 1 antibiotics-12-00010-f001:**
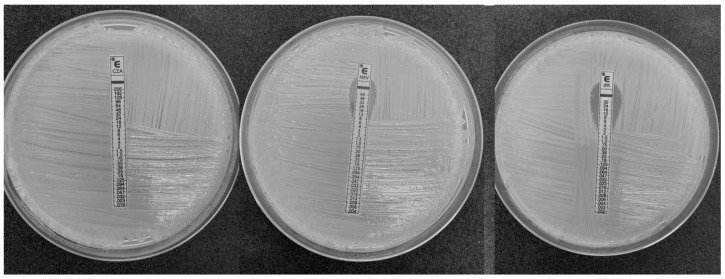
Results of E-test for β-lactam/ β-lactamase inhibitor combinations.

**Table 1 antibiotics-12-00010-t001:** Antimicrobial susceptibility test of cr-Kp.

Antimicrobials	MIC (mg/L)
Amoxicillin/clavulanate	≥32
Piperacillin/tazobactam	≥128
Cefepime	≥32
Cefotaxime	≥64
Ceftazidime	≥64
Ceftolozane/tazobactam	≥32
Meropenem	≥16
Imipenem	8
Amikacin	32
Gentamicin	≥16
Tobramycin	≥32
Ciprofloxacin	≥4
Trimethoprim/sulfamethoxazole	≥320
Ceftazidime/avibactam	256 *
Imipenem/relebactam	3 *
Meropenem/vaborbactam	12 *

* MIC assessed by E-test^®^.

**Table 2 antibiotics-12-00010-t002:** Antimicrobial resistance genes found by Whole Genome Sequencing.

Antibiotics	Resistance Genes
Aminoglycosides	*aac(6’)-Ib4*
	*aadA* *
	*aadA2* ^
	*aph(3’)-Ia*
	*strA.v1* ^
	*strB.v1*
Fluoroquinolones	*qnrS1*
Macrolides	*mphA*
Chloramphenicol	*catA1* ^
	*cmlA5* *
	*floR.v2* *
Rifampicin	*arr-2*
Sulfonamides	*sul1*
	*sul2*
Tetracyclines	*tet(A).v2*
Trimethoprim	*dfrA12*
	*dfrA14.v2* *

* Inexact nucleotide and inexact amino acid match; ^ Inexact nucleotide but exact amino acid match.

**Table 3 antibiotics-12-00010-t003:** Beta-lactamases identified by Whole Genome Sequencing.

Β-Lactamases	
Class A	KPC-31
	TEM-1D
	SHV-11
Class C	CMY-16
Class D	OXA-181
	OXA-10

## Data Availability

The data is available when it is requested for motivated reasons.

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
