# Peer review of "Genomic Characterization of KPC-31 and OXA-181 Klebsiella pneumoniae Resistant to New Generation of β-Lactam/β-Lactamase Inhibitor Combinations"

_antibiotics, 2022, doi:10.3390/antibiotics12010010_

Round 1

Reviewer 1 Report

It is a concise and important paper on characterisation of Carbapenem resistant Klebsiella pneumoniae strain. The study indicates importance of further research for novel combinations of therapeutics for treatment of diseases caused by multi-resistant strains. The article is well written and there are only several issues that should be evaluated. After addressing these issues I recommend it for publishing in your Journal.

Page 1, Lines 44-45. It is generally accepted that the AMR is a global threat, and that there is a strong need to work on every possible field to minimize the problem (including urgent actions and de3velopment of new therapeutic options). But for stating „with an increasing mortality in the next decades“ the authors should include either relevant data or consider rewriting the sentence (for example: using present perfect tense or specific adjectives/adverbs).

Page 5, Line 160 „de novo“ should be written italic? Please proof-read the manuscript.

Author Response

R1: It is a concise and important paper on characterisation of Carbapenem resistant Klebsiella pneumoniae strain. The study indicates importance of further research for novel combinations of therapeutics for treatment of diseases caused by multi-resistant strains. The article is well written and there are only several issues that should be evaluated. After addressing these issues I recommend it for publishing in your Journal.

AA: We thank the Reviewer for having provided an important contribution to the improvement of our manuscript. Following his/her suggestions, besides the next point by point responses, we directly reported several changes in the text, including in the introduction section.

R1: Page 1, Lines 44-45. It is generally accepted that the AMR is a global threat, and that there is a strong need to work on every possible field to minimize the problem (including urgent actions and development of new therapeutic options). But for stating „with an increasing mortality in the next decades“, the authors should include either relevant data or consider rewriting the sentence (for example: using present perfect tense or specific adjectives/adverbs).

AA: We thank the Reviewer for this important comment. We changed the sentence, including relevant data, as follows: “Antimicrobial resistance (AMR) is a major public health threat associated with an increasing mortality in the next decades, it has been estimated that the number of related deaths could rise to 10 million people by 2050”.

R1: Page 5, Line 160 „de novo“ should be written italic? Please proof-read the manuscript.

We thank the Reviewer for this recommendation. The text was revised and edited accordingly.

Reviewer 2 Report

Dear Editor, Thanks for the privilege given to review this manuscript. The manuscript presented a multi-resistant carbapenem-resistant Klebsiella pneumonia which they isolated from a covid patient. The research report is okay as a short communication but the author could have done better by elaborating a little on the method used in the experiment. 

Author Response

R2: Dear Editor, Thanks for the privilege given to review this manuscript. The manuscript presented a multi-resistant carbapenem-resistant Klebsiella pneumonia which they isolated from a covid patient. The research report is okay as a short communication but the author could have done better by elaborating a little on the method used in the experiment. 

AA: We thank the Reviewer for this important comment. The section of the method was edited as follows: “Cr-Kp was detected in a patient admitted to the intensive care unit of an Italian university hospital with a diagnosis of COVID-19-related pneumonia. A bronchoalveolar lavage specimen was collected and used for further analysis. The automated system VITEK II [22] was used to perform strain identification and antibiotic susceptibility testing; the following antibiotics were tested: amoxicillin/clavulanate, piperacillin/tazobactam, cefepime, cefotaxime, ceftazidime, ceftolozane/tazobactam, meropenem, imipenem, amikacin, gentamicin, tobramycin, ciprofloxacin, trimethoprim/sulfamethoxazole. Additionally, in order to perform a specific evaluation of MIC values for CZA, M/V, and I/R, Etest gradient strips (bioMérieux, Inc., Durham, NC) were used. A 0.5 McFarland suspension in 0.85% sterile saline was inoculated to Mueller-Hinton agar plates with a sterile cotton swab and Etest strips were applied to plates manually. After overnight incubation at 37°C for 16-20 h, plates were read. Results were interpreted according to the clinical breakpoints of the European Union Committee on Antimicrobial Susceptibility testing, EUCAST v.11.0 [23]. Carbapenemases encoding genes (i.e., blaKPC, blaNDM, blaVIM, blaIMP, blaOXA-48 like) were detected by real time-PCR [24].

Whole Genome Sequencing was performed to identify genetic determinants responsible for antimicrobial resistance and virulence. After an overnight culture of the strain, genomic DNA was extracted using a genomic DNA extraction kit (QIAamp DNA Kit, QIAGEN, Hilden, Germany) and sequenced using the Illumina Nova Seq 6000 platform (Illumina, San Diego, CA, United States). Raw reads were trimmed for adapters and filtered for quality (average quality > 20) with Fastp (v0.20.1) [25] and quality checked after trimming with FastQC (v0.11.9) [26]. A de novo genome assembly was performed using SPAdes Genome Assembler (v3.14.1) [27], using the ‘-careful’ option; quality of the assembly was evaluated by Quast (v5.1) [28]. Annotation of the assem-bled contigs was performed with Prokka (v1.14.6) [29] and isolate typing was per-formed with mlst tool (v2.11) [30,31]. Kaptive (v0.7.3) and Kleborate (v2.2.0) were used to predict the polysaccharide capsular (K) and lipopolysaccharide O antigen (O) loci, and antimicrobial resistance determinants and virulence factors, respectively [32-34]. MobileElementFinder tool (v1.0.3) [35] and PlasmidFinder (v2.0.1) [36] were used for mobile genetic elements and plasmids identification, respectively. The presence of Tn4401 transposons, their type and flanking sequences were detected by TE-Typer (v1.1) [37].”

Reviewer 3 Report

Shorten methods section if possible 

in the discussion, there is paragraph of 2 lines. Consider combining these with other paragraphs 

Author Response

Shorten methods section if possible 

in the discussion, there is paragraph of 2 lines. Consider combining these with other paragraphs 

AA: We thank the Reviewer for his/her comments and the contribution provided to the improvement of the manuscript. A point-by-point replies were reported and the text modified accordingly with his/her suggestions.

R3:” Shorten methods section if possible”

AA: We thank the reviewer for this comment. We tried to describe all the methods used in our work as briefly as possible. Following both recommendations of Reviewer #2 and #3, the methods section was edited accordingly.

R3:” in the discussion, there is paragraph of 2 lines. Consider combining these with other paragraphs”

AA: We thank the Reviewer for his/her suggestion. The paragraph was modified as follows: “The strain was found to belong to ST512, which is a variant originated from Clonal Group (CG) 258. ST512 is one of the most prevalent high risk Sequence Type detected worldwide and characterized by high transmissibility in nosocomial settings. The rapid and efficient dissemination of KPC has mostly been caused by the clonal expansion of CG 258 (e.g., ST512), as previously reported in national surveys [11,12].”

Round 2

Reviewer 2 Report

The authors have made the suggested corrections